# Determination, Quality, and Health Assessment of Pesticide Residues in Kumquat in China

**DOI:** 10.3390/foods12183423

**Published:** 2023-09-14

**Authors:** Yaohai Zhang, Zhixia Li, Bining Jiao, Qiyang Zhao, Chengqiu Wang, Yongliang Cui, Yue He, Jing Li

**Affiliations:** 1Citrus Research Institute, Southwest University, Chongqing 400712, China; lizhixia@cric.cn (Z.L.); jiaobining@cric.cn (B.J.); zhaoqiyang@cric.cn (Q.Z.); wangchengqiu@cric.cn (C.W.); cuiyongliang@cric.cn (Y.C.); heyue@cric.cn (Y.H.); lijing@cric.cn (J.L.); 2Quality Supervision and Testing Center for Citrus and Seedling, Ministry of Agriculture and Rural Affairs, Chongqing 400712, China; 3Key Laboratory of Quality and Safety Control of Citrus Fruits, Ministry of Agriculture and Rural Affairs, Chongqing 400712, China; 4Laboratory of Quality and Safety Risk Assessment for Citrus Products, Ministry of Agriculture and Rural Affairs, Chongqing 400712, China; 5National Citrus Engineering Research Center, Chongqing 400712, China

**Keywords:** kumquat, pesticide residue, risk assessment

## Abstract

Pesticide residues in kumquat fruits from China, and the quality and chronic/acute intake risks in Chinese consumers, were assessed using the QuEChERS procedure and UHPLC-MS/MS and GC-MS/MS methods. Our 5-year monitoring and survey showed 90% of the 573 samples of kumquat fruits collected from two main production areas contained one or multiple residual pesticides. Overall, 30 pesticides were detected, including 16 insecticides, 7 fungicides, 5 acaricides, and 2 plant growth modulators, of which 2 pesticides were already banned. Two or more residual pesticides were discovered in 81% of the samples, and pesticide residues in 9.4% of the samples surpassed the *MRLs*, such as profenofos, bifenthrin, triazophos, avermectin, spirodiclofen, difenoconazole, and methidathion. The major risk factors on the safety of kumquat fruits before 2019 were profenofos, bifenthrin, and triazophos, but their over-standard frequencies significantly declined after 2019, which was credited to the stricter supervision and management policies by local governments. Despite the high detection rates and multi-residue occurrence of pesticides in kumquat fruits, about 81% of the samples were assessed as qualified. Moreover, the accumulative chronic diet risk determined from *ADI* is very low. To better protect the health of customers, we shall formulate stricter organic phosphorus pesticide control measures and stricter use guidelines, especially for methidathion, triazophos, chlorpyrifos, and profenofos. This study provides potential data for the design of kumquat fruit quality and safety control guidelines and for the reduction in health risks to humans.

## 1. Introduction

Citrus fruits are one of the major commodity fruits worldwide and rank first among all fruits in terms of yields. The global annual trading amount of citrus fruits ranks only after wheat and corn, which makes citrus the third-largest international commercial agricultural product worldwide [1]. Citrus fruits are among the predominant agricultural products of China, which has the largest planting area and yield in the world. There are five main varieties of citrus in China, including loose-skin mandarin, sweet oranges, pomelos, lemons, and kumquats. Kumquat, belonging to *Fortunella*, is a relative of citrus and both of them belong to Rutaceae. Kumquat originated from South Asia, being planted for over 1600 years in the Asia-Pacific and grown worldwide [2]. The main varieties of kumquat include Jindan (*F. crassifolia*), Luofu (*F. margarita*), Luowen (*F. japonica*), Jindou (*F. hindsill*), and other kumquats (*intergeneric hybrids*) according to the Records of Chinese Fruit Trees—Kumquat Fruits [3,4]. Different from most citrus fruits, whole fruits of kumquat are edible, with an intense sweet start and a slightly bitter finish. The annual production of kumquats in 2019 was about 600,000 tons in China, and the total planting area was nearly 390,000,000 m^2^, making China rank first in terms of both yield and planting area. In 2022, Yangshuo and Rong’an of Guangxi accounted for 99% of total yield in China.

Kumquat fruits contain various nutrients and trace elements necessary for the human body, such as vitamins, amino acids, sugars, minerals, pectins, and dietary fibers. Most importantly, kumquat fruits have unique nutritional functional components, which are different from other citrus varieties [5,6,7]. The kumquat fruit planting areas of China are mainly located at the subtropics with north latitudes 22~33° and altitudes below 800 m. The climate in these areas is dominated by high temperature, rains, warmth, and wetness. Together with the long growth period of kumquat fruits and risks of diverse pests and hazards, the quality and security issues of kumquat fruits, including pesticide residues, are always the concern of governments and the public.

Quality and safety are major topics of agricultural products, and are decided by pesticide residues and other risk factors. For this reason, pesticide residues need to be monitored and relevant management measures are taken to prevent food chain pollution [8,9]. Moreover, years of monitoring results indicate that understanding the changing trends of pesticide residues will effectively guide managers in future control works. Currently, supervisory institutions and organizations in many countries provide laws and provisions to regulate and detect the use of pesticides [10]. In China, the Ministry of Agriculture and Rural Affairs is in charge of organizing and implementing pesticide residue evaluation and monitoring. To make sure whether agricultural products have risks for customers, the national security institution recently monitored the diet intake risks of pesticide residues [11,12,13].

The risks in dietary intake of pesticide residues were computed using the consumption and pollution data from national monitoring projects and combining these with international criteria [14]. Then, the result of intake risk was compared with the toxicological reference 74 value (usually the acceptable daily intake or acute reference dose, namely *ADI* or *ARfD*) 75 to validate whether residues in foods are in accord with the corresponding maximum residue 76 limit (*MRL*) [15]. To protect customer health and the environment, the *MRLs* of pesticide residues in foods are provided according to Chinese legislation GB 2763 [16]. So far, there is little research on pesticide residue monitoring and quality security assessment in oranges in China [17,18].

We collected 2922 samples of mandarins and oranges between 2013 and 2018, and detected pesticide residues using the QuEChERS procedure and UHPLC-MS/MS, GC-MS, and GC methods to evaluate the dietary risks to Chinese customers [17]. Results showed that the top risk factors were isocarbophos, triazophos, and carbofuran before 2015, and were gradually dominated by profenofos and bifenthrin after 2016. Chronic dietary risks are acceptable to both general adults and children and will not affect health. Moreover, we detected 16 common insecticides and acaricides in 1633 specimens of oranges (including 261 samples of kumquat fruits from nine varieties) using the QuEChERS procedure and UHPLC-MS/MS and GC-MS/MS methods to systematically analyze the potential health risks of the residues [18]. Results show the safety ranks as lemons > pomelos > ponkan > satsuma mandarin > oranges > citrus hybrids > ‘nanfengmiju’ mandarin > ‘shatangju’ mandarin > kumquat. However, triazophos in all varieties caused acute diet risks to customers, and bifenthrin in ‘nanfengmiju’ mandarin caused acute diet risk to children, which are both unacceptable.

As China is the world’s largest kumquat fruit producer, it is necessary to determine the actual status of kumquat fruits in China for the sake of effective production, supervision, and safe consumption. This study aimed to (i) analyze pesticide residue levels in kumquat fruits of China and the temporal variations of pesticides that exceed their *MRLs*, (ii) evaluate the overall product quality of kumquat fruits in China using the index of quality for residues (*IqR*), and (iii) assess whether the intake levels pose a long-term health risk to the local consumers.

## 2. Materials and Methods

### 2.1. Chemicals and Standards

Overall, 89 types of forbidden, severely restricted, and common pesticides in citrus production were chosen for detecting in the state-wide detection system, including acephate, acetamiprid, aldicarb, amitraz, avermectin, azoxystrobin, bifenthrin, boscalid, bromopropylate, buprofezin, carbendazim, carbofuran, carbosulfan, chlordimeform, chlorfluazuron, chlorothalonil, chlorpyrifos, clothianidin, coumaphos, cyhalothrin, cypermethrin, deltamethrin, 2,4-dichlorophenoxyacetic acid (2,4-D), dichlorvos, dicofol, difenoconazole, diflubenzuron, dimethoate, dipterex, o,p’-DDT, p,p’-DDD, p,p’-DDE, p,p’-DDT, emamectin, fenamiphos, fenitrothion, fenpropathrin, fenpyroximate, fenthion, fenvalerate, fipronil, fludioxonil, flusilazole, fonofos, forchlorfenuron, α-HCB, β-HCB, γ-HCB, δ-HCB, hexythiazox, imazalil, imibenconazole, imidacloprid, isazofos, isocarbophos, isofenphos-methyl, kresoxim-methyl, malathion, metalaxyl, methamidophos, methidathion, methomyl, monocrotophos, myclobutanil, 1-naphthaleneacetic acid, omethoate, paclobutrazol, parathion, parathion-methyl, permethrin, phenothiocarb, phenthoate, phorate, phosmet, phosphamidon, phoxim, pirimicarb, posfolan-methyl, posfolan-methyl, prochloraz, profenofos, propargite, propiconazole, pyridaben, pyrimethanil, quinalphos, spirodiclofen, sulfotep, tebuconazole, terbufos, thiabendazole, thiophanate-methyl, triadimefon, triazophos, and trifloxystrobin. Standard individual chemicals were bought from Dr. Ehrenstorfer GmbH (Augsburg, Germany) and the Environmental Quality Supervision and Testing Center, Ministry of Agriculture (Tianjin, China). HPLC-grade methanol and acetonitrile were from CNW (Augsburg, Germany). HPLC-level acetone and formic acid were from Kelong Chemcial Reagent Co. Ltd. (Chengdou, China). Anhydrous MgSO_4_ and NaCl were at analytical grade (Sinopharm Chemcial Reagent Co. Ltd., Shanghai, China). Primary secondary amine (PSA) sorbent was from CNW (40–63 μm, 6 nm, Germany).

Stock solutions (1000 mg·L^−1^) of individual pesticides were prepared in acetone, n-hexane, or methanol and kept in a brown glass storage bottle at −50 °C until used. The solutions were fully stable for about one year. Standard working solutions at 10 mg·L^−1^ were made by diluting the stock solution into acetone for GC-MS/MS and into acetonitrile for LC-MS/MS. Accordingly, matrix-matched standard solutions at 10–2000 μg·L^−1^ were made by adding blank sample extracts to each diluted standard solution. All water used here was deionized water (18 MΏ cm) from a Milli-Q Advantage A10 SP reagent water device (Millipore, MA, USA).

### 2.2. Apparatuses

A Shimadzu Nexis 2030 gas chromatograph equipped with a programmed split/splitless injector and an AOC 6000 multifunction autosampler (Shimadzu, Kyoto, Japan), and a Shimadzu 8040 138 NX tandem mass spectrometry (Shimadzu, Kyoto, Japan) were used to perform gas chromatography coupled with tandem mass spectrometry (GC-MS/MS) confirmation. An SH-Rxi-5Sil MS (30 m 140 × 0.25 mm id × 0.25 μm film) capillary column was used. A 1290 Infinity UHPLC system was linked to a 6495 Triple Quadrupole LC-MS/MS device added with a jet stream EI source (Agilent, Santa Clara, CA, USA). Data were acquired and analyzed on an Agilent MassHunter Workstation B.07.00. Chromatographic isolation was finished on an Agilent ZORBAX Eclipse Plus C_18_ column (50 mm × 2.1 mm, 1.8 μm) with gradient elution.

Samples were prepared using a GENIUS 3 vortex agitator (IKA, Stauffen, Germany), a CL31R multispeed refrigerated centrifuge (Thermo Scientific, Waltham, MA, USA), a WD12 water bath nitrogen blowing instrument (Aosheng Instrument, Hangzhou, China), and a CK2000 high-throughput tissue grinder (Thmorgan Biotechnology, Beijing, China).

### 2.3. Design of Sampling Plan

The sampling plan for kumquat fruit detection included two parts. First, fruits that will involve the major commodities on the main producing areas were sampled. Hence, kumquat fruits were mainly chosen from Yangshuo and Rong’an of Guangxi province. Second, sufficient pesticide choosing was ensured, and three sets of pesticides were considered: commonly used ones in kumquat fruit growth, newly registered ones for kumquat, and non-compliant ones with *MRLs* or prominent contributors to the Chinese dietary pesticide intake as per the detected results from the last years.

### 2.4. Sampling

From 2016 to 2020, 573 ripe kumquat fruit samples in total were obtained (Figure 1). The sample sources were mostly from plantations and professional cooperatives, and a few from markets as per the official directive process on sampling. Sediments were removed via homogenization before extraction. All kumquat fruits (3 kg each) were in the form of whole fruits. A typical part of the samples (200 g each) was chopped and homogenized in a food chopper. The homogenized samples were stored in sealed polyethylene bottles at −20 °C. The frozen samples were immediately moved to our laboratory using sealed containers with enough ice and kept frozen until tested within one month.

### 2.5. Sample Treatment

Extraction was performed as per the QuEChERS procedure with appropriate modification and optimization. The QuEChERS procedure was elaborated below. (1) A sample (10.00 ± 0.01 g) was placed into a 50 mL FEP centrifuge tube. (2) Acetonitrile (10.00 mL) was added into each tube and oscillated heavily for 1 min. (3) The tubes were kept in a refrigerator at −20 °C for at least 15 min. (4) Anhydrous MgSO_4_ (4.0 g) and 1.0 g of NaCl were added, shaken heavily for 1 min, and (5) centrifuged at 10,000 rpm for 5 min. (6) Extracts (upper layer; 3.00 mL) were decanted into the centrifuge tube with 50 mg of PSA and 300 mg of anhydrous MgSO_4_. (7) The tubes were well capped, vortexed for 1 min, and (8) centrifuged at 4000 rpm for 5 min. (9) Extracts (upper layer; 1.00 mL) were removed into a centrifuge tube, concentrated in a N_2_ stream at 45 °C to dryness, redissolved in 1.00 mL of acetone, and filtered via a 0.22 μm membrane filter for GC-MS/MS. (10) The residues were filtered in the same way.

### 2.6. Instrumental Analysis

Fifty-two pesticides were detected using GC-MS/MS. The carrier gas was helium (≥99.999%) at a flow rate of 1.0 mL/min. The injector was maintained at 250 °C. Injection was performed in the pulse splitless mode. The injection volume was 1.0 μL. The column temperature program was for 60 °C at 1 min, heating first at 40 °C/min to 180 °C, and then at 8 °C/min to 280 °C, and holding for 8 min. The MS setting was as follows: data acquisition in the electron impact (EI) mode at a 70 eV voltage under the multi-reaction monitoring (MRM) mode, transmission line and ion source temperatures at 280 and 200 °C, respectively, and solvent delay time of 3 min. The MS parameters for the 52 pesticides in GC-MS/MS are shown in Table 1.

The other 37 pesticides were monitored via UHPLC-MS/MS. Mobile phase A was water containing 0.1% formic acid (*v*/*v*). Mobile phase B was methanol. The gradient was started with 10% phase B, rose slowly to 90% in 0.2–6 min, from 90% to 98% in 6–9 min, from 98% to 2% in 9–12 min, and then dropped to 10%. The column was kept at 40 °C. The flow rate was 0.3 mL/min, and the injection volume was 3.0 mL. MS conditions were as follows: the EI interface with an Agilent jet stream was adopted in both negative and positive ion modes. Analysis was performed in MRM in a single run. The temperature and flow were 150 °C and 14 L/min in the drying gas, and were 375 °C and 12 L/min in the sheath gas. The nebulizer pressure was 207 kpa (30 psi). The capillary, nozzle, and fragmentor voltages were 4000, 500, and 380 V, respectively. The MS parameters for the 37 pesticides in LC-MS/MS are shown in Table 1.

### 2.7. Quality Control and Assurance

The pesticide residues were quantified using external standard calibration curves. The sensitivity, linearity, precision, and accuracy of the new method were verified. Linearity was acceptable when the multi-level calibration curve (5–500 μg/L) in the linear response interval of the detector for quantification exhibited correlations r^2^ > 0.99. Sensitivity was assessed using the limit of detection (LOD) and limit of quantification (LOQ), which were computed as the lowest dose signal-to-noise (*S*/*N*) ratios of 3 and 10, respectively, by injecting spiked fruit samples. The LOD and LOQ of the method were 2–20 and 10–50 μg·kg^−1^ for the target substances. For precision and accuracy, the recovery rates of three spiked levels (0.01, 0.05, 0.2 mg·kg^−1^) were within 70–130% and thus met the criteria. Six recovery tests were repeated at each spiked level. The relative standard deviations (RSDs) were below 10%, so the repeatability was acceptable.

### 2.8. Index of Quality for Residues (IqR)

*IqR* was computed to test how the monitored levels of multiple pesticides impacted the total quality of the samples. *IqR* for each sample was determined as the sum of the ratio of each pesticide concentration to the *MRL* (Equation (1)). The *MRLs* were cited from the Chinese National Food Safety Standard GB 2763 [16]. The citrus fruits were separated by *IqR* into 4 quality classes: Inadequate (*IqR* > 1.0), Adequate (0.6–1.0), Good (0–0.6), and Excellent (0) [19].
(1)$$IqR=\sum_{i=1}^{n}PRCi/MRLi$$
where *i* is the given pesticide in each sample; *PRC_i_* is the pesticide residue concentration of *i* and *MRL_i_* is the *MRL* of pesticide *i* (both mg·kg^−1^).

### 2.9. Dietary Risk Evaluation

#### 2.9.1. Chronic Intake Risk

The national estimated daily intake (*NEDI*, mg·kg^−1^·bw) and the chronic exposure risk (*%ADI*) of a pesticide were computed as follows:(2)$$NEDI=\frac{STMR\times Fi}{bw}$$
(3)$$\%ADI=\frac{NEDI}{ADI}\times \;100\%$$
where *STMR* (mg·kg^−1^) is the median residue from supervised trials (herein the monitored average residues of pesticides in kumquat samples were used), *F_i_* (kg/d) is the average fruit consumption, *bw* (kg) is the average body weight, and *ADI* (mg·kg^−1^ bw) is the acceptable daily intake of pesticide. The *F_i_* of kumquats by Chinese is 1.96 g/person/d, and the *bw* for the general population (>1 yrs) and children aged 1–6 years is 53.23 and 16.14 kg, respectively [20]. The *ADIs* of the studied pesticides were acquired from GB 2763 [16]. The *%ADI* > 100 and <100 imply chronic risk is unacceptable and acceptable, respectively.

#### 2.9.2. Acute Intake Risk

The international estimated short-term intake (*IESTI*, mg·kg^−1^·bw) and acute exposure risk (*%ARfD*) were determined from Equations (4)–(7). According to the WHO, three types of the equations (Cases 1, 2a, 2b) were used for different commodities [14].

Case 1. (*Ue* < 25 g):(4)$$IESTI=\frac{LP\times HR}{bw}$$

Case 2a. (25 g ≤ *Ue* < *LP*):(5)$$IESTI=\frac{Ue\times HR\times v+(LP-Ue)\times HR}{bw}$$

Case 2b. (*Ue* > *LP*)
(6)$$IESTI=\frac{LP\times HR\times v}{bw}$$
(7)$$\%\mathrm{ARfD}=\frac{IESTI}{\mathrm{ARfD}}\times \;100\%$$
where *LP* (kg) is the large portion, *HR* (mg·kg^−1^) is the highest residue in samples, *Ue* (kg) is the unit weight of the edible portion, and v is variability. Cases 1, 2a, and 2b were used for kumquat; orange and mandarin cultivars; lemon and pummelo, respectively. The data of *LP*, *Ue* and *v* for each commodity are listed in Ref. [18]. The *ARfDs* of the tested pesticides were cited from the WHO database [21]. Similarly, *%ARfD* < 100 and *%ARfD* > 100 reflect acceptable and unacceptable acute risk, respectively.

## 3. Results

### 3.1. Detection of Pesticide Residues

The results of pesticide residues detected with UHPLC-MS/MS and GC-MS/MS are listed in Table 2. Among the 573 samples of kumquat fruits, 30 of the 89 targeted pesticides were accumulatively detected. For each sample, the result of pesticide residues was the average value of three repeated measurements. As per Chinese national standards, GB/T 6379.1 [22] and GB/T 6379.2 [23], precise data were obtained using UHPLC-MS/MS and GC-MS/MS. The reproducibility and repeatability of these methods were determined at 95% reliability. Based on the whole data, 16 insecticides (53.3%), 7 fungicides (23.3%), 5 acaricides (16.7%), and 2 plant growth modulators (6.7%) were identified. The insecticides mainly covered three types: organic phosphorus, pyrethroids, and nicotines (75% together). These pesticides are widely used in kumquat planting in China to prevent and control severe plant diseases such as citrus anthracnose, citrus canker, citrus scab, citrus melanose, and insect pests such as red spiders, bed bugs, stink bugs, scale insects, leaf miners, leafroller moths, and beetles [24,25].

The highest detection rates were found for tebuconazole and spirodiclofen (both 37.9%), followed by profenofos (35.1%), cyhalothrin (32.1%), difenoconazole (25.1%), imidacloprid (24.8%), thiophanate-methyl (24.1%), chlorpyrifos (21.3%), prochloraz (17.8%), propargite (16.9%), carbendazol (15.4%), and hexythiazox (15.2%). These data basically accord with other studies. For instance, chlorpyrifos and carbendazol are the most commonly identified [19,26,27], and the detection rates of prochloraz, tebuconazole, and acetamiprid are very high [28,29]. The high identification rates of spirodiclofen, profenofos, propargite, and hexythiazox (15–38%) are due to the common occurrence of red spiders in kumquat fruits.

In all the samples (N = 573), 9.8% of the kumquat samples were found with no pesticide residues, and 90.2% of the samples had at least one (of the 30 identified pesticides) that exceeded the quantitative limits. The concentrations of the 30 detected residual pesticides ranged from 0.01 to 2.24 mg·kg^−1^. The pesticides at high concentrations (mg·kg^−1^) were propargite (2.25), profenofos (2.10), thiophanate-methyl (1.49), prochloraz (1.10), difenoconazole (1.09), and triazophos (1.01). These pesticide residues were mostly fungicides and insecticides, followed by acaricides. The high concentrations may be ascribed to the wide use before and after harvesting [24]. Nevertheless, the high residue concentrations of these pesticides do not all exceed the *MRLs* of China [16]. The *MRLs* restrict the types and concentrations of pesticides in oranges, indicating pesticides were applied basically in accordance with Good Agricultural Practices.

### 3.2. Pesticide Residues Over-Standardness, and Detection of Banned and Restricted Pesticides

The residue levels of 7 pesticides in the kumquat samples in the 5 tested years surpassed the Chinese *MRLs* [16]. The order ranked by over-standard rate is profenofos (5.24%) > bifenthrin (1.22%) > triazophos (0.70%) > avermectin (0.70%) > spirodiclofen (0.52%) > difenoconazole (0.17%) > methidathion (0.17%). The disqualified pesticides are all insecticides (except for spirodiclofen and difenoconazole), which may be related to the massive use of insecticides to control the frequently occurring insect pests in citrus trees. The detection rates of the two banned or restricted pesticides (methidathion and carbofuran) were 0.5% and 0.3%, respectively. The *MRL* is not a toxicological limit but shall be toxicologically acceptable. Exceeding the *MRL* and the use of forbidden pesticides are both representative of violating GAP. Notably, registration of methidathion to be used in kumquat fruit trees and other vegetables or fruits was canceled by the Ministry of Agriculture of China in 2015 because of its high toxicity. The MRL of methidathion in kumquat fruits was lowered from 2 [30] to 0.05 mg·kg^−1^ [16].

Figure 2 shows the single-residue concentrations of the 30 identified pesticides and 7 over-standard pesticides, which were distributed in 45 of the kumquat samples, in which the residues of at least one pesticide exceeded the *MRL* (accounting for 7.8% of all samples). Avermectin was the most over-standard (1325%*MRL*), followed by profenofos (1051%*MRL*), methidathion (839%*MRL*), bifenthrin (648%*MRL*), triazophos (506%*MRL*), difenoconazole (182%*MRL*), and spirodiclofen (139%*MRL*). In our previous study, the over-standard rate of pear samples collected from China was 2.6%, and the over-standard rates of cyfluthrin, difenoconazole, omethoate, profenofos, pyrimethanil, and tebuconazole were 123–332% [31]. The over-standard rate of peach samples from China was 3.2%, and the over-standard rates of carbendazol, cyhalothrin, cypermethrin, deltamethrin, difenoconazole, fenbuconazole, flusilazole, and isazofos were 104–345% [13]. Among the samples of mandarins and oranges from China, the over-standard rate was 3.8%, and the over-standard rates of bifenthrin, profenofos, fenpyroximate, carbofuran, triazophos, isocarbophos, difenoconazole, and cyhalothrin were 186–283% [17]. Reports show pesticide residues in fruit samples exceed the *MRLs* in other countries or organizations. For instance, Mac Loughlin et al. found that in the 135 samples of fruits and vegetables in Argentina markets, the largest residual pesticide content was detected in oranges—30% of the tested oranges exceeded *MRLs* [19]. In 11% of the tested orange samples in Mexico, the concentrations of methyl chlorpyrifos, malathion, and methidathion were all over the *MRLs* of the European Union [32].

### 3.3. Multi-Residue Pesticide Residues

In the positive samples, 54 samples (9.4%) were found with one pesticide, and 463 samples (80.8%) had two or more pesticides. One sample was found with up to 12 pesticides. The detection rate changing with the number of residual pesticides maximized at four, and then gradually decreased (Figure 3). Multi-residues are ubiquitous in kumquat fruits and other fruits of many countries. Reports from the European Food Safety Authority showed that multiple pesticides were discovered in 58.7% of orange specimens. The largest number of residual pesticides (12) was found in a third country, but some oranges in the European Union contained up to 11 residual pesticides [33]. Poulsen et al. found oranges more frequently contained multiple residues than 20 other types of fruits, and multiple pesticides were detected in 75% of samples, including 93% of mandarin samples, 86% of grapefruit samples, 82% of orange samples, 79% of lemon samples, and 61% of pomelo samples [34]. When multiple pesticides are applied to treat different plant diseases and insect pests, fruits are more susceptible to multi-pesticide residual pollution.

Though the amounts and types of multi-residues vary along with the planting sites, years, and planters, general rules have been discovered. Sampling and testing of fruits and vegetables in markets of Argentina showed chlorpyrifos, endosulfan, and at least one pyrethroid and one fungicide (chlorpyrifos + endosulfan + pyrethroid + fungicide) co-existed in the same sample. The most common pesticide residue combinations in orange samples were acetamiprid + chlorpyrifos + prochloraz + carbendazol (oranges); chlorpyrifos + prochloraz + etoxazole + tebuconazole, and profenofos + acetamiprid + chlorpyrifos + spirotetramat (mandarins) [17]. Moreover, multi-residue pesticides included tebuconazole + spirodiclofen + thiophanate-methyl + prochloraz, and spirodiclofen + profenofos + cyhalothrin + imidacloprid.

### 3.4. Changes of Over-Standard Pesticide Residues in Five Years

The temporal changes of over-standard frequency in seven over-standard residual pesticides are shown in Figure 4, including profenofos (Figure 4a), bifenthrin (Figure 4b), triazophos (Figure 4c), avermectin (Figure 4d), spirodiclofen (Figure 4e), difenoconazole (Figure 4f), and methidathion (Figure 4g). The major risk factors affecting the safety of kumquat fruits before 2019 were profenofos, bifenthrin, and triazophos. After 2019, however, the over-standard frequencies of profenofos, bifenthrin, and triazophos significantly declined, and the banned or restricted pesticides including methidathion were not over-standard, which was credited to the stricter supervision and management policies by local governments. During 2016–2018, the over-standard frequency of profenofos in kumquat fruits slightly increased, which is basically consistent with a previous report that the over-standard frequencies of profenofos in samples of mandarins and oranges steadily increased year by year [17]. Profenofos is easily over-standard in kumquats, which can be explained by three reasons. First, profenofos is registered in only a few products of oranges, and can efficiently prevent and cure red spiders. Currently, profenofos products that are not registered in oranges must have been extensively and illegally used in oranges. Second, farmers use pesticides irregularly, and may increase application doses and times as well as using pesticides at the late mature stage. Third, the residue decomposition rate of profenofos in oranges is associated with the variety and producing environment. Profenofos is a slowly degrading pesticide with a safe period over 60 days. Thus, reasonable use of profenofos and popularization of its substitutes is of concern. As high-risk pesticides are effectively controlled, the over-standard rate of kumquat fruits significantly drop year after year, and the over-standard pesticides are also random. The unqualified rate of kumquat fruits in 2020 slightly rose from that in 2019. One main reason was that GB 2763-2020 modified the limits of some pesticides used to kumquat fruits, which was a ‘decreased dose’ for most pesticides [35]. Thus, irregular pesticide use will increase the quality and safety risks of kumquat fruits. For instance, the limit of avermectin in kumquat fruits provided in GB 2763-2020 dropped from 0.02 to 0.01 mg·kg^−1^, and the limit of spirodiclofen decreased from 0.5 to 0.4 mg·kg^−1^ [35]. Thus, the changes in the limits of pesticides in kumquat fruits provided in GB 2763-2020 shall be further popularized so as to normalize the application of spirodiclofen, avermectin, and other pesticides.

### 3.5. Quality Assessment of Kumquat Fruits

The quality safety of agricultural products in terms of pesticide residues is assessed using *MRLs* worldwide. The multiple residues existing in a single specimen will impact the quality of the whole product through the accumulation or synergistic effect of single residues. *IqR* is an effective indicator to measure the overall quality of foods [24]. In the present study, the quality in the majority of samples is satisfactory (Table 3). Clearly, 9.8%, 58.8%, and 12.4% of the samples were rated as excellent (*IqR* = 0), good (0–0.6), and adequate (0.6–1.0), respectively. The remaining 19.0% of the specimens were inadequate (*IqR* > 1.0), but this proportion is obviously higher than those reported in mandarins and oranges from China [17]. Because of the standardization of relevant data, *IqR* allows for simple and objective comparison. The above data indicate future orchard gardeners can formulate more targeted schemes to modulate the quality of orange fruits in these producing areas. Among the unqualified categories, the pesticide residues in 45 samples (41.3%) exceeded the *MRLs*, and the accumulative pesticide residues of 64 samples (58.7%) reached or were lower than the *MRLs*. Of the 109 unqualified samples, the *IqR* of 71 samples (65.1%) varied between 1 and 2. In other words, the decline in product quality at the accumulative level exceeded 1 to 2 times of the *MRL*. The *IqR* of 34/109 (31.2%) samples was between 2 and 5, and the *IqR* of 4 (3.7%) samples exceeded 5. The *MRL* of the least qualified sample was 22.8 times the appropriate level, and is higher than the reported level [17]. Admittedly, the huge differences among varieties also make the results incomparable. Nevertheless, the accumulative risks of the pesticide residues below the *MRLs* shall also be concerned, as they largely contribute to *IqR*.

### 3.6. Health Risks of Pesticide Residues

The *%ADI* and *%ARfD* were obtained as shown in the Section 2.9. As for accumulative chronic risk assessment, the total of *%ADI* is 0.76 among general people (>1 year old) and 2.50 in children (1–6 years old), which are both smaller than 100 (Table 4). The exposure value of each pesticide is significantly below *ADI*, and the *%ADI* of each pesticide is far below 100. These data indicate the chronic risk of exposure to pesticides via eating kumquat fruits can be ignored. The *%ADI* maximized in methidathion (0.59 in children, 0.18 in general people), followed by avermectin (0.52 and 0.16, respectively) and triazophos (0.45 in children, 0.14 in general people). Generally, among the 30 detected pesticides, the residues of five organic phosphorus pesticides contributed most largely to *%ADI* (44.3%). In particular, the top four organic phosphorus pesticides were methidathion, triazophos, chlorpyrifos, and profenofos, and their total contribution was 99.9% of all organic phosphorus pesticides. The contribution rates of the other 4 insecticides (avermectin, carbofuran, bifenthrin, and buprofezin), the other 8 insecticides, 7 fungicides, and 7 pesticides were 30.5%, 4.3%, 10.4%, and 10.5%, respectively (Figure 5). Similar conclusions were made in other studies from China, Poland, and Brazil that eating fruits will not cause health risks to adults or children [17,36,37]. As for acute risk assessment, the exposure values of all pesticides are significantly lower than *ARfD* except for triazophos, and the *%ARfD* of each pesticide is far lower than 100. The *%ARfD* is the highest for triazophos (212.98 in children, 227.92 in general people), which makes up to an 89% contribution to the total of *%ARfD*. According to the above results, the use of pesticides including methidathion and carbofuran shall be more strictly controlled and punished. More importantly, high-risk organic phosphorus pesticides including triazophos, chlorpyrifos, and profenofos shall be controlled with appropriate measures or lower limits, or be completely forbidden.

The uncertainty of diet risk assessment often originates from toxicological or consumptive data, processing factors, left censored data processing, and loss of exposure assessment model [27]. Jensen et al. found the intake exposure of pesticides by humans was changeable, and was mainly decided by degradation in crops, harvesting time, and processing [38]. Clearly, processing factors are practically uncertain in assessing diet risks of primary agricultural products. Generally, pesticide residues of fruits and vegetables can be lowered by washing, soaking, peeling, blanching, or other domestic processing [39]. Moreover, the diet exposure to pesticides is reduced by 3% to 11.5% after washing, machinery, or thermal processing [40]. Here, we ignored the processing factors during diet risk assessment, so the results of exposure may be overestimated. Thus, more precise and real exposure estimation shall be made in the future.

## 4. Conclusions

Our 5-year monitoring and survey showed 90% of the 573 samples of kumquat fruits collected from two main production areas contained one or multiple residual pesticides. Overall, 30 pesticides were detected, including 16 insecticides, 7 fungicides, 5 acaricides, and 2 plant growth modulators, of which 2 pesticides were already banned. The commonly detected pesticides include tebuconazole, spirodiclofen, profenofos, cyhalothrin, difenoconazole, and imidacloprid. Two or more residual pesticides were discovered in 81% of the specimens, and pesticide residues surpassed the *MRLs* in 9.4% of the samples, including profenofos, bifenthrin, triazophos, avermectin, spirodiclofen, difenoconazole, and methidathion. The largest over-standard rate of 1325% *MRL* was found in avermectin. Profenofos, bifenthrin, and triazophos were the main safety risk factors of kumquat fruits before 2019, but their over-standard frequencies significantly declined after 2019, indicating the over-standard pesticides were random to some extent. Despite the high detection rates and multi-residue occurrence of pesticide residues in kumquat fruits, about 81% of the samples were assessed as qualified. Moreover, the accumulative chronic diet risk determined from *ADI* is very low. To better protect the health of customers, we shall formulate stricter organic phosphorus pesticide control measures and stricter use guidelines, especially for methidathion, triazophos, chlorpyrifos, and profenofos. This study provides potential data for the design of kumquat fruit quality and safety control guidelines and for the reduction in health risks to humans. In addition, our study performed in this manuscript improves on others performed for citrus fruits by the same authors [17,18].

## Figures and Tables

**Figure 1 foods-12-03423-f001:**
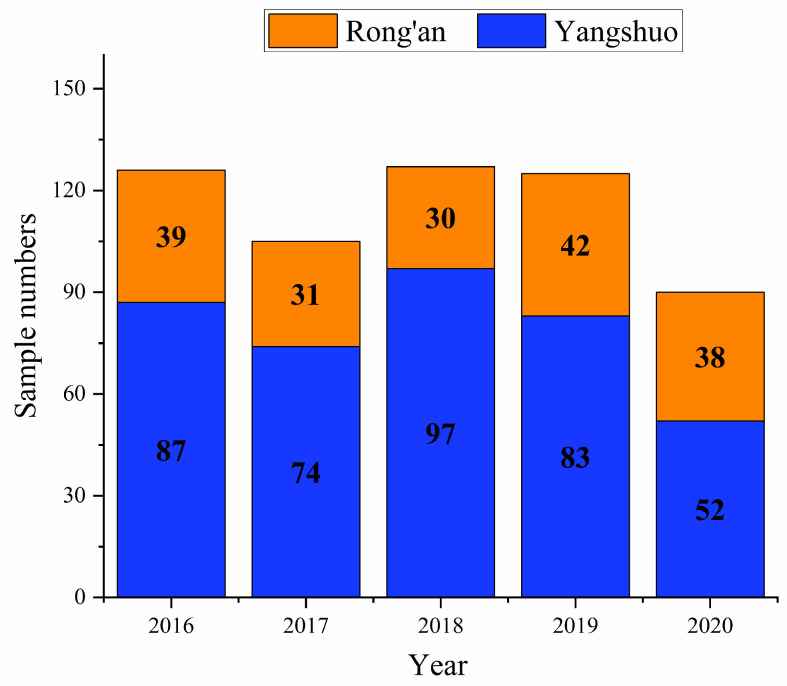
Number of kumquat fruit samples tested during 2016–2020.

**Figure 2 foods-12-03423-f002:**
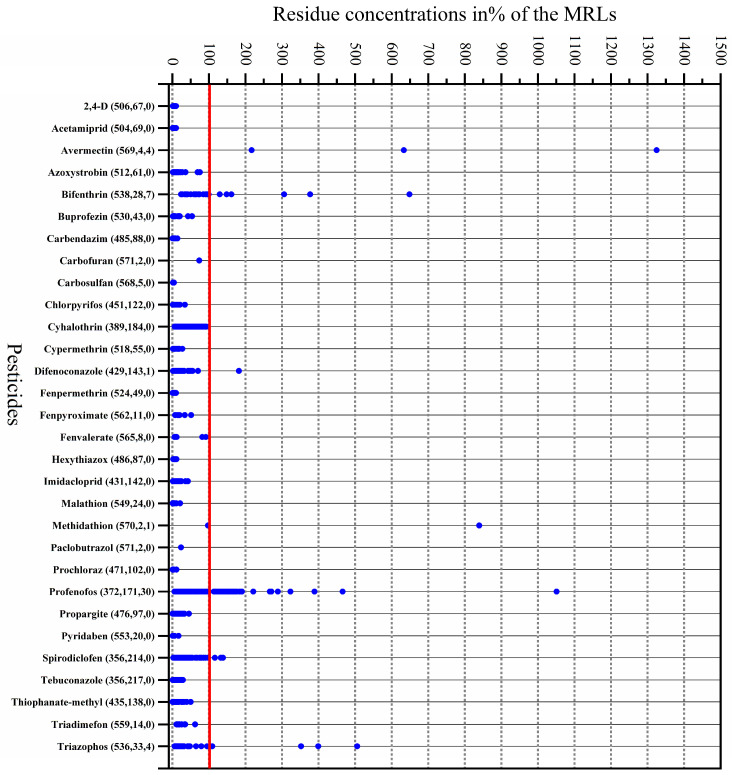
Residue contents of the thirty detected pesticides in kumquat samples, expressed as percentage of the *MRL* (numbers in brackets after the pesticide name refer to the numbers of samples below the LOQ, above the LOQ, below the *MRL*, and above the *MRL*. The blue dot represents the distribution of pesticide residue content and the red line represents MRL).

**Figure 3 foods-12-03423-f003:**
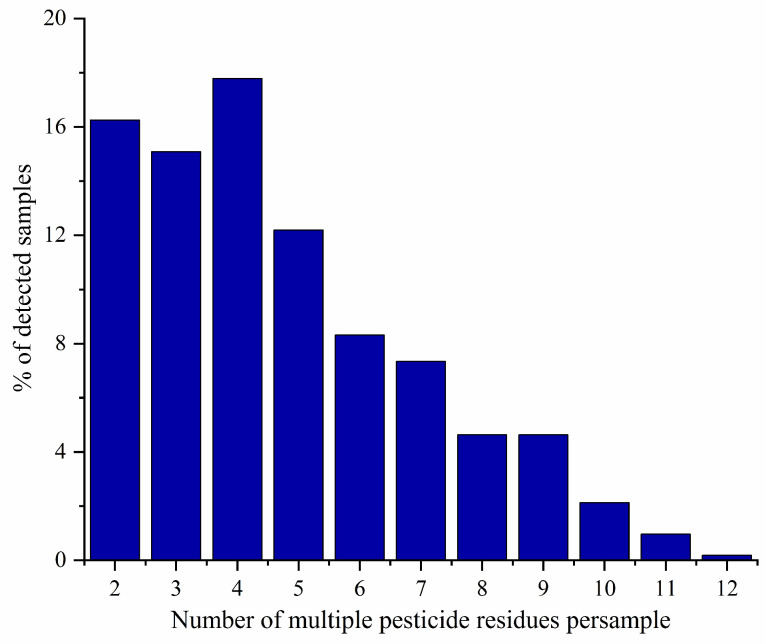
Proportion of multi-pesticide residues in kumquat fruit samples.

**Figure 4 foods-12-03423-f004:**
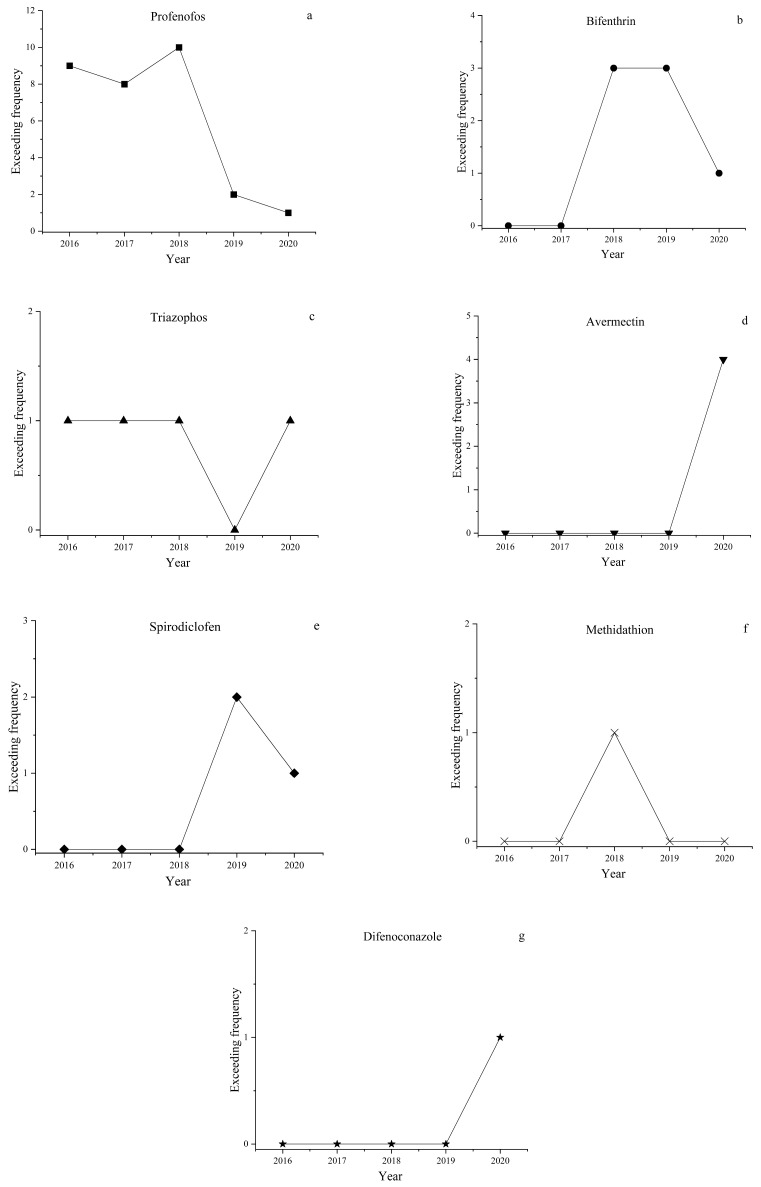
Temporal changes of *MRL* surpassing frequency in the seven pesticides from 2016 to 2020 ((**a**): profenofos; (**b**): bifenthrin; (**c**): triazophos; (**d**): avermectin; (**e**): spirodiclofen; (**f**): difenoconazole; (**g**): methidathion).

**Figure 5 foods-12-03423-f005:**
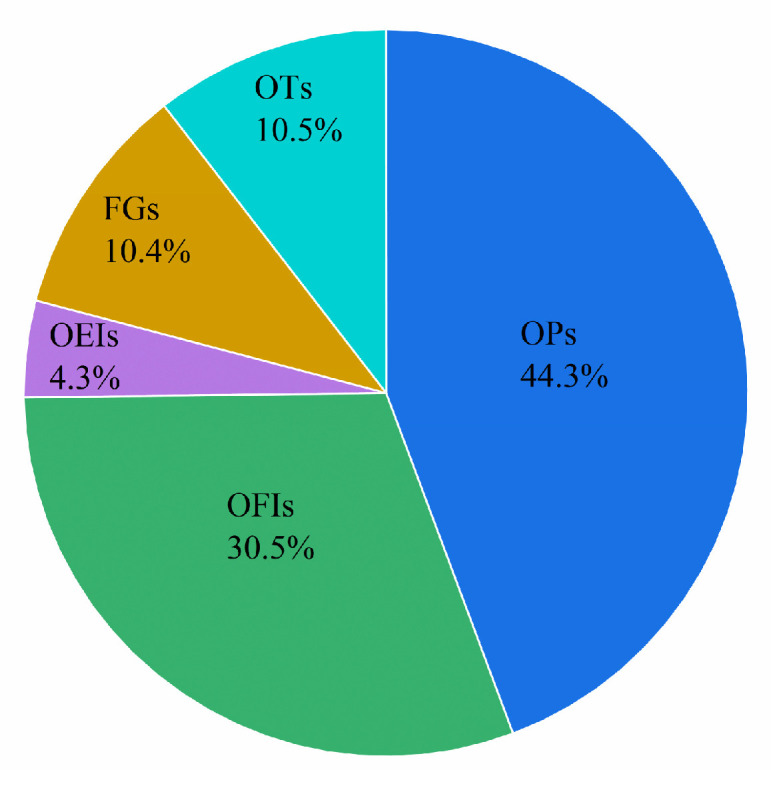
Contributions of the five classes of 30 detected pesticides to hazard index. OPs = organophosphorus pesticides, OFIs = other four insecticide pesticides, OEIs = other eight insecticide pesticides, FGs = fungicide pesticides, OTs = other pesticides.

**Table 1 foods-12-03423-t001:** The MS parameters for the 89 pesticides in GC-MS/MS and LC-MS/MS detection.

Analyte	Method	Quantitative Ion (*m*/*z*)	Qualitative Ion (*m*/*z*)	CE (eV)
bifenthrin	GC-MS/MS	181.1 > 166.1	181.1 > 179.1	12; 12
bromopropylate	GC-MS/MS	340.9 > 182.9	340.9 > 184.9	18; 20
buprofezin	GC-MS/MS	172.1 > 57.0	175.1 > 132.1	14; 12
chlordimeform	GC-MS/MS	196.0 > 181.0	181.0 > 140.0	10; 15
chlorothalonil	GC-MS/MS	263.9 > 168.0	263.9 > 228.8	24; 18
chlorpyrifos	GC-MS/MS	196.9 > 168.9	313.9 > 257.9	14; 14
coumaphos	GC-MS/MS	362.0 > 109.0	362.0 > 226.0	16; 14
cyhalothrin	GC-MS/MS	181.1 > 152.1	163.1 > 91.0	24; 22
cypermethrin	GC-MS/MS	163.1 > 127.1	163.1 > 91.0	6; 14
deltamethrin	GC-MS/MS	180.9 > 151.9	252.9 > 93.0	22; 20
dicofol	GC-MS/MS	139.0 > 111.0	139.0 > 75.0	16; 28
difenoconazole	GC-MS/MS	323.0 > 265.0	265.0 > 202.0	14; 20
dimethoate	GC-MS/MS	125.0 > 47.0	125.0 > 79.0	14; 8
o,p’-DDT	GC-MS/MS	235.0 > 165.0	237.0 > 165.0	24; 28
p,p’-DDD	GC-MS/MS	235.0 > 165.0	237.0 > 165.0	24; 28
p,p’-DDE	GC-MS/MS	246.0 > 176.0	317.9 > 248.0	30; 24
p,p’-DDT	GC-MS/MS	235.0 > 165.0	237.0 > 165.0	24; 28
fenamiphos	GC-MS/MS	303.1 > 195.1	288.1 > 260.1	8; 6
fenitrothion	GC-MS/MS	277.0 > 260.0	277.0 > 109.1	6; 14
fenpropathrin	GC-MS/MS	181.1 > 152.1	265.1 > 210.1	22; 12
fenpyroximate	GC-MS/MS	213.0 > 77.1	213.0 > 168.5	24; 24
fenthion	GC-MS/MS	278.0 > 109.0	278.0 > 169.0	20; 14
fenvalerate	GC-MS/MS	225.1 > 119.1	225.1 > 147.1	20; 10
fludioxonil	GC-MS/MS	248.0 > 127.0	248.0 > 154.0	26; 20
fonofos	GC-MS/MS	137.1 > 109.1	246.0 > 137.1	8; 6
α-HCB	GC-MS/MS	180.9 > 144.9	218.9 > 182.	16; 8
β-HCB	GC-MS/MS	180.9 > 144.9	218.9 > 182.	16; 8
γ-HCB	GC-MS/MS	180.9 > 144.9	218.9 > 182.	16; 8
δ-HCB	GC-MS/MS	180.9 > 144.9	218.9 > 182.	16; 8
hexythiazox	GC-MS/MS	184.0 > 149.0	156.0 > 112.0	10; 15
imazalil	GC-MS/MS	215.0 > 173.0	215.0 > 159.0	6; 6
isazofos	GC-MS/MS	257.0 > 162.0	257.0 > 119.0	8; 18
isocarbophos	GC-MS/MS	289.1 > 136.0	230.0 > 212.0	14; 10
isofenphos-methyl	GC-MS/MS	199.0 > 121.0	241.1 > 121.1	14; 22
malathion	GC-MS/MS	173.1 > 99.0	173.1 > 127.0	14; 6
methidathion	GC-MS/MS	145.0 > 85.0	145.0 > 58.0	8; 14
monocrotophos	GC-MS/MS	127.1 > 109.0	127.1 > 95.0	12; 16
omethoate	GC-MS/MS	156.0 > 110.0	110.0 > 79.0	8; 10
parathion	GC-MS/MS	139.0 > 109.0	291.1 > 109.0	8; 14
parathion-methyl	GC-MS/MS	263.0 > 109.0	125.0 > 47.0	14; 12
permethrin	GC-MS/MS	183.1 > 153.1	183.1 > 168.1	14; 14
phenothiocarb	GC-MS/MS	160.1 > 72.0	160.1 > 106.1	10; 12
phenthoate	GC-MS/MS	273.9 > 125.0	273.9 > 246.0	20; 6
phorate	GC-MS/MS	260.0 > 75.0	231.0 > 129.0	8; 24
phosmet	GC-MS/MS	160.0 > 77.0	160.0 > 133.0	24; 14
phosphamidon	GC-MS/MS	127.1 > 109.1	127.1 > 95.1	12; 18
pirimicarb	GC-MS/MS	238.1 > 166.1	166.1 > 55.0	12; 20
posfolan-methyl	GC-MS/MS	168.0 > 109.0	168.0 > 136.0	15; 15
posfolan-methyl	GC-MS/MS	255.0 > 227.0	255.0 > 140.0	6; 22
profenofos	GC-MS/MS	338.9 > 268.9	336.9 > 266.9	18; 14
propargite	GC-MS/MS	135.1 > 107.1	135.1 > 77.0	16; 24
pyridaben	GC-MS/MS	147.1 > 117.1	147.1 > 132.1	22; 14
pyrimethanil	GC-MS/MS	198.1 > 183.1	198.1 > 118.1	14; 28
quinalphos	GC-MS/MS	146.1 > 118.0	146.1 > 91.0	10; 24
sulfotep	GC-MS/MS	322.0 > 202.0	322.0 > 174.0	10; 18
terbufos	GC-MS/MS	231.0 > 128.9	231.0 > 174.9	26; 14
triadimefon	GC-MS/MS	208.1 > 181.0	208.1 > 111.0	10; 22
triazophos	GC-MS/MS	161.0 > 134.0	161.0 > 106.0	8; 14
acephate	LC-MS/MS	184.0 > 143.0	184.0 > 49.0	20; 20
acetamiprid	LC-MS/MS	223.1 > 126.0	223.1 > 90.0	27; 45
aldicarb	LC-MS/MS	213.2 > 115.9	213.2 > 88.6	30; 25
amitraz	LC-MS/MS	294.2 > 163.1	294.2 > 122.1	30; 35
avermectin	LC-MS/MS	896.6 > 752.5	896.6 > 449.4	50; 55
azoxystrobin	LC-MS/MS	404.1 > 372.1	404.1 > 329.1	8; 32
boscalid	LC-MS/MS	343.0 > 307.0	343.0 > 272.0	16; 32
carbendazim	LC-MS/MS	192.1 > 160.1	192.1 > 132.1	16; 32
carbofuran	LC-MS/MS	222.1 > 165.1	222.1 > 123.1	20; 30
carbosulfan	LC-MS/MS	381.2 > 160.2	381.2 > 118.1	12; 36
chlorfluazuron	LC-MS/MS	539.9 > 383.0	539.9 > 158.0	44; 36
clothianidin	LC-MS/MS	250.0 > 169.0	250.0 > 132.0	12; 20
2,4-dichlorophenoxyacetic acid	LC-MS/MS	219.0 > 161.0	221.0 > 163.0	15; 15
dichlorvos	LC-MS/MS	221.0 > 127.0	221.0 > 109.0	27; 23
diflubenzuron	LC-MS/MS	311.0 > 158.0	311.0 > 141.0	8; 32
dipterex	LC-MS/MS	256.9 > 220.9	256.9 > 108.9	4; 15
emamectin	LC-MS/MS	886.5 > 158.0	886.5 > 82.1	40; 60
fipronil	LC-MS/MS	435.0 > 330.0	435.0 > 250.0	12; 28
flusilazole	LC-MS/MS	316.1 > 165.0	316.1 > 247.0	24; 12
forchlorfenuron	LC-MS/MS	248.1 > 129.0	248.1 > 93.0	22; 44
imibenconazole	LC-MS/MS	413.2 > 125.9	413.2 > 170.9	30; 20
imidacloprid	LC-MS/MS	256.0 > 208.9	256.0 > 175.0	12; 12
kresoxim-methyl	LC-MS/MS	314.2 > 222.1	314.2 > 267.0	10; 0
metalaxyl	LC-MS/MS	280.2 > 160.1	280.2 > 220.1	20; 10
methamidophos	LC-MS/MS	142.1 > 125.0	142.1 > 107.1	4; 3
methomyl	LC-MS/MS	163.1 > 106.0	163.1 > 88.0	4; 0
myclobutanil	LC-MS/MS	289.1 > 125.1	289.1 > 70.1	32; 16
1-naphthaleneacetic acid	LC-MS/MS	185.0 > 141.1	185.0 > 141.1	4; 4
paclobutrazol	LC-MS/MS	294.1 > 125.2	294.1 > 70.1	36; 16
phoxim	LC-MS/MS	299.0 > 129.1	299.0 > 77.1	4; 24
prochloraz	LC-MS/MS	376.0 > 265.9	376.0 > 308.0	12; 4
propiconazole	LC-MS/MS	342.1 > 159.0	342.1 > 69.1	32; 16
spirodiclofen	LC-MS/MS	411.1 > 313.0	411.1 > 71.2	5; 15
tebuconazole	LC-MS/MS	308.1 > 125.0	308.1 > 70.0	47; 40
thiabendazole	LC-MS/MS	202.0 > 175.0	202.0 > 131.0	24; 36
thiophanate-Methyl	LC-MS/MS	343.0 > 151.0	343.0 > 93.0	20; 56
trifloxystrobin	LC-MS/MS	409.1 > 145.0	409.1 > 186.0	52; 12

CE: collision energy.

**Table 2 foods-12-03423-t002:** Occurrence of pesticide residues in kumquat fruits of China.

Pesticide	Type	No. (%) of Positive Samples	Concentration Range (mg·kg^−1^)	Mean Valve (mg·kg^−1^)	No. (%) of Exceedance	*MRL* (mg·kg^−1^)
2,4-D	P	67 (11.7)	0.010–0.096	0.033		1
acetamiprid	I	69 (12.0)	0.010–0.194	0.042		2
avermectin	I	4 (0.7)	0.016–0.132	0.058	4 (0.70)	0.01
azoxystrobin	F	61 (10.6)	0.011–0.751	0.098		1
bifenthrin	I	35 (6.1)	0.011–0.324	0.051	7 (1.22)	0.05
buprofezin	I	43 (7.5)	0.010–0.538	0.087		1
carbendazim	F	88 (15.4)	0.010–0.665	0.068		5
carbofuran	I	2 (0.3)	0.011–0.015	0.013		0.02
carbosulfan	I	5 (0.9)	0.011–0.041	0.023		1
chlorpyrifos	I	122 (21.3)	0.010–0.340	0.051		1
cyhalothrin	I	184 (32.1)	0.010–0.185	0.052		0.2
cypermethrin	I	55 (9.6)	0.011–0.273	0.042		1
difenoconazole	F	144 (25.1)	0.010–1.092	0.077	1 (0.17)	0.6
fenpermethrin	I	49 (8.6)	0.011–0.474	0.093		5
fenpyroximate	A	11 (1.9)	0.018–0.256	0.086		0.5
fenvalerate	I	8 (1.4)	0.011–0.183	0.055		0.2
hexythiazox	A	87 (15.2)	0.010–0.055	0.020		0.5
imidacloprid	I	142 (24.8)	0.010–0.423	0.040		1
malathion	I	24 (4.2)	0.012–0.416	0.071		2
methidathion	I	3 (0.5)	0.010–0.420	0.160	1 (0.17)	0.05
paclobutrazol	P	2 (0.3)	0.013–0.116	0.064		0.5
prochloraz	F	102 (17.8)	0.011–1.104	0.099		10
profenofos	I	201 (35.1)	0.011–2.102	0.119	30 (5.24)	0.2
propargite	A	97 (16.9)	0.010–2.248	0.257		5
pyridaben	A	20 (3.5)	0.010–0.332	0.045		2
spirodiclofen	A	217 (37.9)	0.010–0.554	0.075	3 (0.52)	0.4
tebuconazole	F	217 (37.9)	0.010–0.570	0.101		2
thiophanate-methyl	F	138 (24.1)	0.010–1.493	0.178		3
triadimefon	F	14 (2.4)	0.066–0.617	0.254		1
triazophos	I	37 (6.5)	0.011–1.011	0.115	4 (0.70)	0.2

**Table 3 foods-12-03423-t003:** Quality evaluation of the analyzed kumquat fruits according to the calculated *IqR* factor.

*IqR*	No. of Samples	%	Quality Categories
0	56	9.8	excellent
0–0.6	337	58.8	good
0.6–1	71	12.4	adequate
>1	109	19.0	inadequate

**Table 4 foods-12-03423-t004:** Health hazard index for pesticide residues in kumquat fruits.

Pesticide	*ADI* (mg·kg^−1^ bw)	*NEDI* (mg·kg^−1^ bw)		*%ADI*		ARfD (mg·kg^−1^ bw)	*IESTI* (mg·kg^−1^ bw)		*%ARfD*	
		Gen Pop, >1 yrs	Children, 1–6 yrs	Gen Pop, >1 yrs	Children, 1–6 yrs		Gen Pop, >1 yrs	Children, 1–6 yrs	Gen Pop, >1 yrs	Children, 1–6 yrs
Chlorpyrifos	0.01	1.26 × 10^−6^	4.15 × 10^−6^	0.0126	0.0415	0.1	7.67 × 10^−4^	7.16 × 10^−4^	0.77	0.72
Malathion	0.3	1.24 × 10^−6^	4.10 × 10^−6^	0.0004	0.0014	2	9.37 × 10^−4^	8.76 × 10^−4^	0.05	0.04
Triazophos	0.001	1.36 × 10^−6^	4.49 × 10^−6^	0.1362	0.4493	0.001	2.28 × 10^−3^	2.13 × 10^−3^	227.92	212.98
Profenofos	0.03	2.11 × 10^−6^	6.96 × 10^−6^	0.0070	0.0232	1	4.74 × 10^−3^	4.43 × 10^−3^	0.47	0.44
Methidathion	0.001	1.79 × 10^−6^	5.91 × 10^−6^	0.1791	0.5908	0.01	9.46 × 10^−4^	8.84 × 10^−4^	9.46	8.84
Triadimefon	0.03	6.96 × 10^−6^	2.30 × 10^−5^	0.0232	0.0765	0.08	1.39 × 10^−3^	1.30 × 10^−3^	1.74	1.62
Cypermethrin	0.02	8.83 × 10^−7^	2.91 × 10^−6^	0.0044	0.0146	0.04	6.16 × 10^−4^	5.76 × 10^−4^	1.54	1.44
Fenvalerate	0.02	6.17 × 10^−7^	2.03 × 10^−6^	0.0031	0.0102	-	4.13 × 10^−4^	3.86 × 10^−4^	-	-
Bifenthrin	0.01	1.37 × 10^−6^	4.51 × 10^−6^	0.0137	0.0451	-	7.30 × 10^−4^	6.83 × 10^−4^	-	-
Cyhalothrin	0.02	1.29 × 10^−6^	4.24 × 10^−6^	0.0064	0.0212	0.02	4.18 × 10^−4^	3.90 × 10^−4^	2.09	1.95
Fenpermethrin	0.03	2.38 × 10^−6^	7.86 × 10^−6^	0.0079	0.0262	-	1.07 × 10^−3^	9.99 × 10^−4^	-	-
Propargite	0.01	1.47 × 10^−6^	4.83 × 10^−6^	0.0147	0.0483	-	5.07 × 10^−3^	4.74 × 10^−3^	-	-
Pyridaben	0.01	8.34 × 10^−7^	2.75 × 10^−6^	0.0083	0.0275	-	7.49 × 10^−4^	6.99 × 10^−4^	-	-
Difenoconazole	0.01	1.78 × 10^−6^	5.87 × 10^−6^	0.0178	0.0587	0.3	2.46 × 10^−3^	2.30 × 10^−3^	0.82	0.77
Fenpyroximate	0.01	2.57 × 10^−6^	8.47 × 10^−6^	0.0257	0.0847	0.02	5.77 × 10^−4^	5.39 × 10^−4^	2.89	2.70
Hexythiazox	0.03	6.33 × 10^−7^	2.09 × 10^−6^	0.0021	0.0070	-	1.25 × 10^−4^	1.16 × 10^−4^	-	-
Buprofezin	0.009	1.19 × 10^−6^	3.92 × 10^−6^	0.0132	0.0436	0.5	1.21 × 10^−3^	1.13 × 10^−3^	0.24	0.23
Tebuconazole	0.03	1.82 × 10^−6^	5.99 × 10^−6^	0.0061	0.0200	-	1.28 × 10^−3^	1.20 × 10^−3^	-	-
Acetamiprid	0.07	9.43 × 10^−7^	3.11 × 10^−6^	0.0013	0.0044	-	4.38 × 10^−4^	4.09 × 10^−4^	-	-
Carbendazim	0.03	1.35 × 10^−6^	4.44 × 10^−6^	0.0045	0.0148	0.5	1.50 × 10^−3^	1.40 × 10^−3^	0.30	-
Imidacloprid	0.06	1.01 × 10^−6^	3.35 × 10^−6^	0.0017	0.0056	0.4	9.53 × 10^−4^	8.91 × 10^−4^	0.24	0.22
Thiophanate-methyl	0.09	3.12 × 10^−6^	1.03 × 10^−5^	0.0035	0.0114	-	3.37 × 10^−3^	3.15 × 10^−3^	-	-
Spirodiclofen	0.01	1.58 × 10^−6^	5.20 × 10^−6^	0.0158	0.0520	-	1.25 × 10^−3^	1.17 × 10^−3^	-	-
Prochloraz	0.01	2.27 × 10^−6^	7.48 × 10^−6^	0.0227	0.0748	0.1	2.49 × 10^−3^	2.32 × 10^−3^	2.49	2.32
Azoxystrobin	0.2	2.36 × 10^−6^	7.80 × 10^−6^	0.0012	0.0039	-	1.69 × 10^−3^	1.58 × 10^−3^	-	-
2,4-D	0.01	9.91 × 10^−7^	3.27 × 10^−6^	0.0099	0.0327	-	2.17 × 10^−4^	2.03 × 10^−4^	-	-
Carbofuran	0.001	4.79 × 10^−7^	1.58 × 10^−6^	0.0479	0.1579	0.001	3.31 × 10^−5^	3.10 × 10^−5^	3.31	3.10
Carbosulfan	0.01	7.88 × 10^−7^	2.60 × 10^−6^	0.0079	0.0260	0.02	9.33 × 10^−5^	8.72 × 10^−5^	0.47	0.44
Avermectin	0.001	1.56 × 10^−6^	5.16 × 10^−6^	0.1564	0.5157	-	2.99 × 10^−4^	2.79 × 10^−4^	-	-
Paclobutrazol	0.1	2.37 × 10^−6^	7.82 × 10^−6^	0.0024	0.0078	0.05	2.62 × 10^−4^	2.45 × 10^−4^	0.52	0.49

Gen pop: General population.

## Data Availability

The data used to support the findings of this study can be made available by the corresponding author upon request.

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
