# Peer review of "Determination, Quality, and Health Assessment of Pesticide Residues in Kumquat in China"

_foods, 2023, doi:10.3390/foods12183423_

Round 1
Reviewer 1 Report
This paper, which deals a study about the determination, temporal variation and potential health risk assessment of pesticide residues in kumquat fruits in China, should be revised in the following respects:
General comments:
· All text: Authors should separate paragraphs with spaces. Tables and figures should be separated with a space from the rest of the text.
· The text of the manuscript should be revised again by a native speaker because several grammatical errors have been observed.
Specific comments:
· Abstract. The authors should include information on the method used for the determination of the pesticides. They should also include information on sensitivity, precision and accuracy of the method.
· Section 1. Lines 81-100: the two paragraphs should be summarized. Authors should include a paragraph indicating the objectives of the research work and how they will be achieved. A paragraph on methodologies for the determination of pesticides should be included. References related to methodologies should be included.
· Section 2.6. The authors should include a table indicating the ions selected for identification and quantification by GC-MS/MS and HPLC-MS/MS. Analytes should be separated according to whether they are determined by GC or HPLC.
· Section 2.7. The authors should include a table with LOD, LOQ, recoveries and precision data for each of the pesticides.
· Table 1. The authors should define “No. (%) of exceed-ance”
· Section 3.4. A table with the analysis results of all pesticides from 2016 to 2019 should be included, not only for seven pesticides as indicated in Figure 4.
· Table 3: It should be explained in a comprehensive manner how the results are achieved.. The abbreviations used in the table should be defined in the table footer. The same in the other tables.
· Conclusions: It should be noted that the study performed in this manuscript improves on others performed for citrus fruits by the same authors [17,18].
· References: The DOI of the articles should be included in the References Section.
The text of the manuscript should be revised again by a native speaker because several grammatical errors have been observed.
Author Response
Thank you very much for taking the time to review this manuscript. Please find the detailed responses below and the corresponding revisions/corrections highlighted/in track changes in the re-submitted files.

Reviewer 2 Report
In all, the paper was interesting to read.
Minor editing of the English would be welcome.
line 48 if the number is the production in 2019; better to say: annual production in 2019
line 49: the area mu is not well known to international readers - please change to ha.
line 51: altitudes
line 52: for the human body
line 57: rains
line 58: it is not clear what is meant with: long periodic biological phenomena - please change
line 65: effectively
line 66: supervisory
line 68: add the (the Ministry)
line 73: and combining these with
line 76: are in accord
line 78: are provided
line 85: for
line 89: contained
line 95: for
line 127: Stock solutions of (delete "the")
line 137: A Shimadzu...
lines 141-143: is not a whole sentence
line 164: were from markets
line 270: are listed
line 285: please delete duplicate The highest detection rates were found in
line 286: for
line 293: is because
None
Author Response

(The authors gave the same response as above.)

Reviewer 3 Report
In the manuscript foods-2505160, the authors presented original research about determination quality and health risk assessment of pesticide residues in kumquat fruits in China during five years monitoring. Authors focus on detection of pesticide residues, included banned, restricted and those over-standards. Moreover, they showed samples with multi-residue pesticide residues. Authors performed quality and health risk assessment of detected pesticides in kumquat fruits and explained changes in over-standard pesticide residues during monitoring. Therefore, I believe the manuscript will interest Foods readers. But, I recommend some modifications, as per the comments/suggestions below.
Title
1. The reviewer recommend change the title on: Determination, quality and health assessment of pesticide residues in kumquat in China
Introduction:
2. Line 49 In planting area use SI units instead of mu use m2.
3. Lines 81-100 should be moved to the discussion section.
4. At the end of the introduction section the aim must be clearly defined.
Materials and methods
5. In which concentration range the method is linear.
6. In line 216 authors defined LOD and LOQ of the method 2-20 mg/kg and 10-50 mg/kg. Make sure you entered the units correctly.
7. Line 243 I was better to use not general population (>1 years) and children (2-6 years) but divide into adults and children. General population include children between 2-6 years.
Results
8. In line 285 is a mistake. ‘The highest detection rates were found in’ this statement should be deleted.
9. In lines 439-440 change ordinary people as you wrote before to general population or change according to the reviewer's recommendations to adults.
Author Response

(The authors gave the same response as above.)

Round 2
Reviewer 1 Report
The manuscript has been improved from the previous version, but the following modifications should be made:
· Section 2.7. Due to the importance of the analytical features of the method, the authors should include a table with LOD, LOQ, recoveries and precision data for each of the pesticides. In addition, a summary of these analytical features should be included in the Abstract of the manuscript as suggested in the previous review.
· As stated in the previous revision of the manuscript, Table 2. The authors should define “No. (%) of exceedance”
· As stated in the previous revision of the manuscript, Section 3.4. A table with the analysis results of all pesticides from 2016 to 2019 should be included, not only for seven pesticides as indicated in Figure 4.
· As stated in the previous revision of the manuscript, the abbreviations used in the Table 4 should be defined in the table footer.
· Conclusions: It should be noted that the study performed in this manuscript improves on others performed for citrus fruits by the same authors [17,18]. In addition, the authors should indicate the advantages of the studies carried out in this manuscript over the other two methods found in the literature [17,18].
Minor editing of English language required
Author Response

(The authors gave the same response as above.)

Reviewer 3 Report
The authors addressed most of my comments. I do not have addtional suggestions.